# Technological Assessment of Bread with the Addition of *Cyperus esculentus* L. Accessions Flour Grown in the Kuban–Azov Plain

**DOI:** 10.3390/foods14213680

**Published:** 2025-10-28

**Authors:** Nina G. Kon’kova, Valentina I. Khoreva, Vitaliy S. Popov, Tamara V. Yakusheva, Ilya A. Kibkalo, Leonid L. Malyshev, Alla E. Solovyеva, Tatiana V. Shelenga

**Affiliations:** Nikolai Ivanovich Vavilov All-Russian Institute of Plant Genetic Resources, Bolshaya Morskaya Street 42-44, 190000 Saint-Petersburg, Russia; horeva43@mail.ru (V.I.K.); v.popov@vir.nw.ru (V.S.P.); kos-vir@yandex.ru (T.V.Y.); i.kibkalo@vir.nw.ru (I.A.K.); l.malyshev@vir.nw.ru (L.L.M.); alsol64@yandex.ru (A.E.S.); t.shelenga@vir.nw.ru (T.V.S.)

**Keywords:** *Cyperus esculentus*, oil content, protein, starch, fiber, phenolic substances, antioxidant capacity, baking qualities, functional food products, composite flour

## Abstract

Due to increased interest in new functional food products, 20 accessions of chufa tubers from the collection of the N.I. Vavilov Institute of Plant Industry, grown in the Kuban–Azov Plain in 2022, as well as bread samples made from mixed flour (70% whole-grain wheat flour, 30% chufa tuber flour) were studied. Biochemical, farinographic, and baking evaluations were carried out. Differences between the properties of dough with the addition of flour from various accessions of chufa tubers were recorded. According to the results of comparative, dispersion, and principal component analysis, all biochemical indicators (oil, fiber, sum of phenolic substances, antioxidant activity) of chufa tuber flour and bread with added chufa flour surpassed control samples (whole-grain wheat flour and wheat bread), with the exception of protein and starch content. Viscoamylographic, farinographic sedimentation, and baking quality evaluations indicated that the dough made from mixed flour was stronger than the control (dough from whole-grain wheat flour), more resistant to kneading, and had a lower degree of liquefaction. In terms of organoleptic properties, differences were also identified, and the accessions that enhance the taste of mixed bread were selected. Therefore, a preliminary conclusion can be drawn that chufa tubers grown in the conditions of the Kuban–Azov Plain with high rheological properties and high sedimentation values of the mixed dough can be recommended for improving the baking properties not only of wheat but also of other bread cereals. Chufa is also a promising crop in the manufacture of functional food products in the Krasnodar region and for the food industry in general as a potential thickener.

## 1. Introduction

*Cyperus esculentus* L. is a valuable agricultural crop that is widespread in Africa (Nigeria, Mali, Senegal, Ghana), Italy (Sicily), and Spain. The largest commercial plantings in Europe are located in Spain, in the province of Valencia [1]. The yield of chufa is quite high; depending on the region of cultivation and the growing method, the tuber yield of chufa varies widely from 3 to 8.5 tons per hectare. In Russia, at the moment, there are no significant commercial plantings; the cultivation of chufa is carried out by individual farmers on small areas.

Chufa is a heat- and moisture-loving plant; for its cultivation, light soils with good drainage are suitable. Chufa possesses remarkable adaptability, which allows obtaining high yields when cultivated on less fertile sandy soils [2]. The vegetation period is 110–120 days. However, our previous work has shown that chufa can also be grown in more northern regions of the world, such as the southwestern and Central-Chernozem regions of the Russian Federation [3]. For food purposes, the tubers that formed on the plant’s roots are used. The tubers contain 20–25% oil, 25–30% starch, 20% carbohydrates, 5–7% protein, and 7–11% fiber [4,5]. The oil also possesses valuable food properties. The most important fatty acid in chufa oil is oleic acid, the content of which reaches more than 70% [6,7]. Chufa oil is characterized by a high content of vitamin E [8]. The high biomass yield and the content of nutrients in chufa tubers make them a promising source of food, oil, and feed to meet the growing needs of the food industry.

Currently, functional foods receive great attention due to their nutritional value. These products contain all the ingredients necessary for a healthy diet and have a beneficial effect on human health [9,10]. Therefore, increasing the use of various food products based on chufa is very promising [11]. Chufa is used in various branches of the food industry; for example, it is widely used in the confectionery industry for the production of chocolate, sweets, cookies, and cupcakes [12,13,14,15,16]. The traditional Spanish drink “Horchata de Chufa” made from chufa tubers is very popular [17,18,19]. Based on “Horchata de Chufa”, many drinks are produced with the addition of soy milk, fruit juices, and extracts from the leaves of various plants [20,21]. Lactic acid bacteria are also used to prepare milk from chufa tubers [22,23].

Due to its high biologically active properties and its content of phenolic compounds, fatty acids, and fiber, chufa flour is used for the production of combined bakery products. Bread prepared from a mixture of wheat flour with chufa tuber flour has good nutritional properties with high dietary fiber content and reduced starch content [24]. It has been shown that chufa is a promising ingredient for the preparation of gluten-free bread [25] and special products for diabetics. The number of people suffering from these diseases is increasing over time [26]. Due to the trends towards climate change and the demand for an expanding the range of functional foods, there is a necessity to investigate the crops that were previously not considered economically significant or typical for production in certain regions of the world [3]. Bread is a staple food in the daily human diet. It is possible to increase the nutritional value and taste of bread using functional components from edible chufa tuber flour. The aim of this research is to investigate 20 accessions of chufa tubers of various origins from the VIR collection for biochemical composition and technological parameters of bread prepared from a mixture of whole-grain wheat flour (70%) and chufa tuber flour (30%). This study will help to confirm that the chufa accessions available in the VIR collection have good potential for creating new functional food products of regional importance (Krasnodar region of the Russian Federation).

## 2. Materials and Methods

### 2.1. Material

#### 2.1.1. Plant Material

Twenty accessions of chufa from the VIR collection (Table 1) were grown at the Kuban Experimental Station (KES VIR 45°13′ N, 40°47′ E, Kuban–Azov plain) in 2022. The Kuban Experimental Station of the All-Russian Institute of Plant Industry is situated in the steppe zone of Krasnodar region (65 m elevation). It features thick-layered Priazov chernozem soils with 130–140 cm humus horizons. The region has a temperate continental climate and variable precipitation, with average annual temperatures of 10.4 °C. The whole-grain flour of the spring soft wheat variety Pamyati Suslyakova from the VIR collection was used for the composite bread preparation and as a control. This variety was previously used as a control for composite bread evaluation [27].

#### 2.1.2. Bread Material

Twenty model samples of bread were made from mixed flour (30% flour from chufa tuber accessions and 70% whole-grain wheat flour). The bread made from 100% whole-grain wheat flour was used as a control.

### 2.2. Biochemical Analysis

The following parameters were investigated: protein content, oil content, starch content, fiber content, total phenolic compounds, and antioxidant activity. The analysis was performed in two biological replicates, and for each replicate, a separate grain aliquot was taken. All parameters were calculated on a dry weight basis.

#### 2.2.1. Preparation of Plant Material for Study

Chufa tubers and bread samples were preliminarily dried at room temperature, then ground using a laboratory disc mill CM 290 Cemotec (FOSS, Höganäs, Sweden) to particles of about 10 μm in size. To prevent changes in the properties of plant material when exposed to light and oxygen, the flour of chufa tubers was stored in metal containers with airtight lids. Before analysis, the flour was thoroughly homogenized, and two portions were selected for each analysis. An experimental procedure was carried out in the Department of Biochemistry and Molecular Biology using methods for screening collection’s accessions adopted at VIR [3].

#### 2.2.2. Protein Content Analysis

Protein/nitrogen content was determined by the Kjeldahl method (conversion factor for chufa—5.5) using an automatic Kjeltec 2200 protein analyzer (FOSS, Höganäs, Sweden); protein content was expressed as % of the dry weight of the sample.

#### 2.2.3. Oil Content Analysis

Oil content—by the mass of dry fat-free residue using a Soxhlet apparatus with petroleum ether (40–70 °C) as solvent; oil content was expressed as % of the dry weight of the sample.

#### 2.2.4. Starch Content Analysis

Starch content—by the Ewers polarimetric method (conversion factor for chufa—203) using an automatic SAC-i polarimeter/saccharimeter (ATAGO, Kobe, Japan); starch content was expressed as % of the dry weight of the sample.

#### 2.2.5. Fiber Content Analysis

Fiber content—by the gravimetric Weende method on a Dosi-Fiber FIWE-6 extraction unit (VELP, Usmate Velate, Italy); fiber content was expressed as % of the dry weight of the sample.

#### 2.2.6. Dry Matter Content (Moisture) Analysis

Dry matter content (moisture) was determined by weighing a portion of the ground material (chufa, wheat flour, and ground material from composite bread and wheat bread) before and after drying at 100–102 °C to constant weight using a Memmert drying oven (Memmert GmbH , Schwabach, Germany); dry matter was expressed as % of the weight of the sample before drying.

#### 2.2.7. Total Phenolic Compounds Analysis

Total phenolic compounds (PC) were determined spectrophotometrically using Folin–Ciocalteu reagent. The results were expressed as mg gallic acid equivalents/100 g sample [26].

#### 2.2.8. Antioxidant Activity Analysis

Antioxidant activity (AOA) was measured spectrophotometrically using DPPH reagent as a free radical donor according to the method [28]. The results were expressed as mg ascorbic acid equivalents/100 g sample.

### 2.3. Technological Assessment

#### 2.3.1. Rheological Assessment

The rheological properties of composite dough were studied using a Brabender Farinograph (Brabender GmbH & Co, Duisburg, Germany).

##### Farinographic Evaluation

Sample Preparation

An amount of 10 g of the flour sample was placed in a mixer of the same volume, which is an integral part of the Brabender Farinograph (Germany). Then, distilled water was added to the flour to achieve the desired water absorption capacity. The dough was then kneaded until it reached a standard consistency (500 units). The registration of the rheological properties of the flour sample continued for 12 min from the beginning of the dough liquefaction or from the beginning of dough consistency drop. The farinographic properties of the dough samples were automatically recorded using a Brabender Farinograph (Germany).

Farinographic Indicators Evaluation

The following indicators were studied: water absorption capacity (WAC) of the dough, dough formation time or dough development time (DDT) and dough stability (DS), dough softening or dilution of the dough (DD), and valorimeter number (VN). WAC characterizes the technological properties of the raw materials—the amount of water necessary to obtain a dough of standard consistency. In the mixed bread samples, WAC substantially decreased. DDT and DS characterize the strength of the flour. It is believed that higher indicators correspond to better quality. In the mixed bread samples, the dough substantially strengthened. DD shows how strongly the weakening of the dough occurs during its prolonged mixing. The higher indicator corresponds to the weaker dough. For the mixed samples, softening significantly decreased. VN—an indicator generalizing the assessment of rheological properties on this device. The higher this value is, the better the quality of the bread sample. Units of farinographic indicators measurement: WAC—%, DDT, DS—min, and DD—device units.

##### Viscoamylographic Assessment

Sample Preparation

A total of 100 mL of distilled water was added to 15 g of flour sample to prepare a homogeneous suspension. The prepared suspension was placed in the measuring module of the micro visco-amylograph device (Brabender GmbH & Co, Duisburg, Germany). The measurements were carried out using a ready-made protocol for potato starch in automatic mode. The measurement time was 20 min. The measurement results were recorded as visco-amylograms.

Viscoamylographic Indicators Evaluation

The study of the starch thermodynamic properties was conducted on a micro visco-amylograph from Brabender (Germany). The essence of the method consists of changing the temperature regime of the suspension of the ground material during constant stirring and recording the load on the paddle. Thus, the viscosity of the paste is measured, and the time and temperature of the onset of a particular phase of gelatinization were recorded. For the tests, a protocol for potato starch was used. The temperature regime consisted of gradually heating the suspension to 92 °C, a holding period at this temperature, gradually cooling the suspension to 50 °C, and a short holding period at this temperature at the end of the test. Thus, two or three thickening peaks were recorded. The protocol was intended for pure starch, where a sample weight of 5 g was assumed. Since our material is not pure starch, we had to increase the sample weight to 15 g. The following viscoamylographic parameters were detected: pasting time (PT), time to peak viscosity (TPV); pasting temperature start (PTS), peak viscosity temperature (PVT); initial pasting viscosity (IPV), peak viscosity in the hot state (PVHS), minimum hot paste viscosity (MHPV), cold paste peak viscosity (CPPV), viscosity at the end of measurement (VEM), viscosity drop at peak temperature (VDPT), viscosity increase upon cooling (VIUC), viscosity increase from hot to cold (VIHC). PT and TPV were measured in min, PTS and PVT in °C, and IPV, PVHS, MHPV, CPPV, VEM, VDPT, VIUC, and VIHC in device units.

#### 2.3.2. Sedimentation Analysis (Swelling Capacity of Flour)

##### Sample Preparation

An amount of 1 g of the flour sample was suspended in graduated test tubes with cork stoppers along with 6 mL of a 2% acetic acid solution in distilled water. After the suspension was allowed to stand for 4–5 min, the volume of the precipitate was measured (first phase). Then, a 1.9% aqueous solution of sodium dodecyl sulfate (SDS) was added to the same sample. The mixture was resuspended, was allowed to stand for 2 min, and the volume of the precipitate was measured (second phase).

##### Sedimentation Indicators Evaluation

The two phases of this methodology—flour swelling capacity phase 1 (PSC1) and flour swelling capacity phase 2 (PSC2)—show the ability of protein structures to resist increasing physicochemical loads or, thanks to them, to realize their swelling potential. Accordingly, the ratio of the indicators of the two phases—sedimentation index or phase 2/phase 1 (P2-1)—characterizes the stability of the protein structures. If it is close to 1, then the sample is stable and has realized its potential. If it is less than 1, the sample is less stable to loads; if it is more than 1, the sample carries greater swelling potential. PSC1 and PSC2 parameters were measured in mL.

#### 2.3.3. Flour Color Evaluation

The color indicators were assessed using a special flour module of the Precision Colorimeter NR20XE, 3nh (Shenzhen, Guangdong China). The color indicators were expressed as parameters: *L* (flour color brightness), *a* (red–green range of the spectrum), *b* (yellow–blue range of the spectrum), *С* (chromaticity index), and *h* (shade angle). The values of *L* are measured in the range from 0 (black) to 100 (white); *a* represents the red color at positive values or green at negative values; *b* represents yellow at positive values or blue at negative values. For a more complete assessment of flour color, the calculation of the hue angle (*h*°) and the chroma index *C*, analogous to saturation or color intensity, are used. They are obtained by calculation from the values *a* and *b*. *C* is calculated as ½ (*a**2 + *b**2) and represents the hypotenuse of a right triangle created by connecting the points (*a*, *b*) and (*a*, 0). The hue angle *h* is calculated as the arctangent of *b*/*a*, the angle between the hypotenuse and the 0° axis (blue–green/red–purple). The values of all color indicators are calculated automatically in accordance with the software of the Precision Colorimeter NR20XE Sw version: 3nh_NR20XE_V00.01 .

#### 2.3.4. Baking Evaluation

##### Bread Sample Preparation

Bread samples were baked from dough made from 15 g of mixed and whole-grain wheat flour. Dough preparation for baking or proofing in a CRT-8-01 proofing cabinet (Abat, Cheboksary, Russia) was carried out at a temperature of 35 °C and a relative humidity of 75–85%. Hearth bread (hereinafter referred to as bread) was baked in a laboratory convection oven KPP-4M (Abat, Cheboksary, Russia) at a temperature of 270 °C for 10–15 min.

##### Baking Indicators Evaluation

One of the main parameters of bread quality is its shape stability (SS) indicator h/d: the ratio of its height to diameter. The higher the indicator, the higher the technological quality of the sample. Another indicator of bread quality, porosity, was not evaluated, as it did not differ significantly between samples. The physical properties of the crumb were evaluated as follows: density (CD) (mass per unit volume), and hardness (CH) using the structural analyzer ST-2 (Quality Laboratory, Moscow , Russia), and the staling of the crumb over 24 h (S24), defined as the increase in sample hardness over 24 h. CD was determined as the ratio of the mass of a cylindrical crumb sample to the volume occupied by this cylinder in g/mL; CH and S24 were measured in g load. Taste quality assessment or taste quality score (TQS) was conducted on a five-point system (1—very bad, 2—bad, 3—fair, 4—good, 5—very good).

### 2.4. Statistical Processing

Statistical processing was carried out using the Statistica 12.0 application software package (Dell Software Company, Palo Alto, CA, USA,StatSoft, Inc., Tulsa, OK, USA (2019), STATISTICA (data analysis software system), version 12. www.statsoft.com (accessed on 1 October 2020) and included the following:

Calculation of basic descriptive statistics for the studied samples (mean, error of mean, coefficient of variation); calculated averages were used for further analysis; analysis of variance to assess the significance of the influence of the “sample” and “experimental treatment” factors; factor analysis of the correlation system of characters.

## 3. Results and Discussion

### 3.1. Biochemical Assessment

Despite the increased interest in new functional food products, there is not much research on products based on chufa, which makes our experiment more relevant. As a control for the chufa flour samples, whole-grain wheat flour was used, and for the samples of bread from mixed flour, bread made only from wheat flour was used. In Figure 1, data on the protein content in the chufa samples and bread are presented. Thanks to the high protein content in wheat flour (21.09%), the samples of wheat bread with chufa also have a higher protein content compared to the samples of chufa tubers. The average protein content in the chufa samples is 6.15%, and in the bread samples—16.21% (bread control—21.68%). The highest protein content was found in the chufa sample k-23 (7.31%) and bread sample k-10 (17.94%), and the lowest content was noted in the chufa sample k-9 (5.41%) and bread k-27 (13.55%). The maximum protein values established for chufa in the current study are slightly higher than we previously established (5.4–6.5%) [3] and were consistent with the data [29]. The protein level in bread made from mixed flour with the addition of 30% chufa, studied by Shaban et al., has lower protein values (8.73%), which is apparently related to the use of lower protein wheat flour for the bread preparation (11.80%) [29]. Our results correspond to the data obtained by Özcan for mixed bread with the addition of 20% chufa flour (15.61%) [24].

The results for starch content in the chufa and bread samples are presented in Figure 2. In this case, a similar situation is observed, and thanks to the high starch content in wheat flour (54.42%), the samples of bread from mixed flour also have a higher starch content compared to the chufa flour samples. The average starch content in the chufa flour samples is 24.53%, and in the bread samples—42.71% (control—54.42%, 52.69% for flour and bread, respectively). The highest starch content was found in the flour of chufa sample k-21 (26.89%) and the bread samples k-7, 9, 14, and 21 (45.69%, 45.68%, 45.33%, and 45.29%, respectively), and the lowest content was noted in the chufa sample k-9 (21.52%) and bread k-13 and 17 (37.19% and 26.08%), which is consistent with our previous data [3].

In Figure 3, data on the oil content in the chufa and bread samples are presented. Thanks to the high oil content in chufa tubers, which significantly exceeds that of wheat flour (2.28%), the bread with chufa addition also has approximately 2.5 times higher oil content than the control bread (2.29%). The average oil content in the chufa flour samples is 22.03%, and in the bread samples, 8.10%. The highest oil content was found in the chufa flour sample k-12 (26.64%) and bread samples k-10, 12, and 27 (9.42, 9.24, and 9.02%, respectively). Conversely, the lowest content was observed in chufa flour k-17 (17.28%) and bread k-9 (6.41%). The results obtained for the 2022 chufa reproductions do not differ significantly from those for 2020–2021 [3]. Shaban et al. note a higher oil content in chufa tubers (28.50%). This is probably due to the conditions of the reproduction region [29]. Other researchers found a higher oil content (16.81%) in bread samples with 30% chufa addition [29], and a lower content (4.90%) in bread with 40% chufa addition [24]. The difference in biochemical parameters is obviously due to the characteristics of the chufa accessions chosen for making bread and the reproduction conditions.

In Figure 4, data on the fiber content in the chufa flour samples and bread samples are presented. Since the chufa flour samples contain approximately two to three times more fiber than the wheat flour control (2.61%), the samples of bread with chufa addition also have an increased fiber content. The fiber content of bread from whole grain wheat flour was almost the same as that of wheat flour. The average fiber content in the chufa flour samples was 7.88%, and in the bread samples, 4.19%. The highest fiber content was found in the chufa flour k-9 (11.11%) and the bread samples k-9, 16, and 27 (5.16, 4.54, and 4.40% respectively), and the lowest content was noted in the chufa flour k-19 (6.95%) and bread k-11 (3.93%) and k-19 (3.91%). Our data obtained for chufa tuber flour is consistent with Shaban et al.’s data (8.50%). However, the amount of fiber in bread with 30% chufa addition was significantly lower (2.43%) [29].

The results on the content of phenolic substances and AOA in the chufa samples and bread are presented in Figure 5 and Figure 6. The average content of phenolic substances and AOA in the chufa flour samples was 125.6 mg% and 19.6 mg%, respectively, and in the bread samples—191.0 mg% and 22.2 mg%, respectively. The highest content of phenolic substances and AOA was found in the chufa flour k-9 (298.1 mg% and 34.7 mg%), and the lowest content of phenolic substances—in flour k-12 (93.4 mg%) and AOA—in k-16 (12.1 mg%). In the bread samples, the highest content of phenolic substances was found in samples k-9, 17, and 24 (213.8, 208.1, and 212.1 mg%, respectively), and AOA—in samples k-2, 9, 19, and 26 (25.6, 24.2, 24.4, and 24.4 mg%, respectively). The lowest content of phenolic substances was detected in samples k-10 and 14 (174.3 and 175.5 mg%) and AOA—in samples k-7 and 13 (20.2 and 19.5 mg%). The amount of phenolic compounds in the chufa tubers selected for baking mixed bread, as reported by Shaban et al. (210.5 mg%), falls within the range of variability in the current study [29]. Ozkan’s PC results for bread with 40% chufa were significantly lower than our results (92.42 mg%), while the AOA results were higher (3.15 mmol/kg or 55.4 mg%) [24].

In terms of all biochemical indicators, except protein and starch, all samples of chufa tuber flour and bread with chufa flour addition surpass control samples: whole-grain wheat flour and wheat bread. Among all the bread samples with chufa tuber flour addition, k-9 stood out with a high fiber content (5.16%), sum of phenolic substances (213.8 mg%), and AOA (24.2 mg%). K-10—high protein and oil content (17.94 and 9.42%, respectively); k-27—oil and fiber content (9.42 and 4.40%, respectively); k-16—high fiber content (4.54%); k-24—high sum of phenolic substances (212.1 mg%); k-19, 26—high AOA (24.2 and 24.4 mg%, respectively). Thus, we can assume that these chufa accessions (k-9, 10, 16, 19, 24, 27) can be used as raw materials for the production of functional food.

### 3.2. Statistical Evaluation of Biochemical Parameters

#### 3.2.1. Comparative Analysis of the Biochemical Parameters

Comparative analysis of the biochemical composition of flour and bread showed that the oil and fiber content of chufa is higher in flour than in bread, while in wheat, these levels are consistent and lower than those of chufa (Table 2, Figure 7A,D). Conversely, the protein and phenolic substance content increases in the bread (Table 2, Figure 7B,E). The starch content in wheat remains at the same level, whereas for chufa, it increases sharply in the bread; overall, starch levels are higher in wheat (Table 2, Figure 7C). Regarding AOA, wheat exhibits low AOA in flour, which increases sharply in bread, while chufa shows stable AOA that is slightly higher in bread compared to wheat (Table 2, Figure 7F).

#### 3.2.2. Correlation Analysis

Correlation analysis of the biochemical composition of chufa tuber flour revealed highly significant positive correlations between oil and protein content (r = 0.728), starch and phenolic compounds (r = 0.767), oil and fiber (r = 0.728), and fiber and AOA (r = 0.775), as well as highly significant negative correlations between oil and starch content (r = −0.826), starch and fiber (r = −0.784) and starch and AOA (r = −0.698). Correlation analysis of the biochemical composition of composite bread showed highly significant positive correlations between phenolic compounds and oil content (r = 0.708), and AOA and oil content (r = 0.952). A highly significant positive correlation was also observed between the fiber content in composite bread and oil content (r = 0.728) and AOA (r = 0.774) in chufa flour, as well as between phenolic compounds in composite bread and AOA in chufa flour (r = 0.722). Additionally, a highly significant negative correlation was found between starch content in chufa flour and fiber content in composite bread (r = −0.783) (Table 3).

#### 3.2.3. Analysis of Variance

The results of the analysis of variance showed that all traits differed significantly among the variants (flour, bread) and (except for phenolic compound content) by the culture factor (pure wheat, chufa and wheat mixture), as well as by the interaction of factors (Variant*Culture) (Table 4).

In terms of protein content, the main share of influence is attributed to the “Variant” factor (Figure 8A); in terms of oil, starch, and fiber content—to the “Culture” factor (Figure 8B–D); in terms of AOA—it is approximately equal for both factors (Figure 8F); and in terms of phenolic compounds—it is mainly associated with unaccounted factors (Figure 8E).

For protein content, the main share of influence is attributed to the “Variant” factor (Figure 8B); for oil, starch, and fiber content—to the “Culture” factor (Figure 8A,C,D); for AOA—it is approximately equal for both factors (Figure 8F); and for PC—it is primarily associated with unaccounted factors (Figure 8E).

#### 3.2.4. Factorial Analysis

Analysis of the system of correlations showed that the factor structure of the variation of traits of the biochemical composition of flour (chufa, wheat) and composite bread (30% chufa/70% wheat) is practically identical (similar) regarding the content of oil, fiber, phenolic substances, and AOA. The following factors were identified. Factor 1—oil content, starch, and fiber vary (chufa flour, wheat flour); oil, protein, and fiber vary (composite bread); Factor 2—phenolic substances and AOA vary (chufa flour, wheat flour) and phenolic substances and AOA vary (composite bread); Factor 3—protein varies (chufa flour, wheat flour) and starch varies (composite bread). АОА—antioxidant activity; B/Oil—oil content of composite bread; B/Pr—protein content of composite bread; B/St—starch content of composite bread; B/F—fiber content of composite bread; B/PC—phenolic compounds content of composite bread; B/АОА—antioxidant activity of composite bread. Regarding the protein content, a positive structure of variation was noted in chufa flour and wheat flour, and a negative one in the composite bread, which indicates a strong influence of thermal processing on the protein content during the baking of bread (Table 5, Figure 9).

In the system of factors identified during the analysis of the composition of flour and bread, pure wheat and samples of composite bread baked from accessions of chufa k-9 and k-17 stand out (Figure 10).

### 3.3. Technological Assessment Indicators

One of the main objectives of the current investigation was to identify differences between various chufa accessions in their baking application. At the preliminary stage of research on the rheology of the mixed dough, these differences were recorded. It was also noted that these differences are most evident when the ratio of wholegrain wheat flour to chufa tuber flour is 70:30%.

#### 3.3.1. Sedimentation Analysis

Sedimentation analysis shows the ability of the experimental material to sustain stable swelling. All major methods of processing cereal crops are based on this ability. In wheat, this indicator is directly linked to the quality of gluten. Thus, its expressiveness is mainly influenced by the properties of the protein—its structural quality, and in some cases to its quantity as well. For the sedimentation analysis, a two-phase methodology recommended for non-wheat crops was used [30,31]. This methodology shows the ability of protein structures to resist increasing physicochemical loads or, thanks to them, to realize their swelling potential. Accordingly, the ratio of the indicators of the two phases characterizes the stability of the protein structures. If it is close to 1, then the sample is stable and has realized its potential. The further the sedimentation value is from 1, the less resistant the sample is to loads; the higher it is compared to 1, the greater swelling potential it carries. Almost all sedimentation values for the flour samples were equal to (k-9, 14, 21, 24), tended towards (k-12, 13, 15, 19, 20, 23), or slightly exceeded 1. The largest part of the samples formed the last group (k-1, 2, 7, 8, 10, 11, 16, 17, 26, 27). The maximum sedimentation index was demonstrated by the flour of the chufa samples k-11 (1.11).

#### 3.3.2. Rheological Assessment Indicators

##### Viscoamylographic Visco-Mylographic

Viscoamylographic indicators allow us to characterize the thermodynamic properties of the carbohydrate complex of various flour samples. During the analysis of chufa tuber flour samples 2 (Figure 11J–M,S) or 3 (Figure 11A–I,N–R,T), thickening peaks were recorded. PT and PTS for all analyzed flour samples were approximately the same and averaged 6.26 min and 77.47 °C. Based on IPV, the flour samples could be divided.

Into groups with a viscosity of less than 80 (k-12, 14, 17, 19, 20, 27, Figure 11H,J,M–O,T) and more than 80 (k-1, 2, 7, 8, 9, 10, 11, 13, 15, 16, 21, 23, 24, 26, Figure 11A–G,I,K,L,P–S) units. The second group turned out to be more numerous. The time of maximum viscosity, or TPV, occurred on average at the ninth minute from the start of the experiment; the difference between the maximum and minimum values for the flour samples was 2.07 min. The temperature at which maximum viscosity or PVT was reached was either close to 90 °C or slightly exceeded this value, averaging 91.86 °C. The difference between the minimum or MHPV (297 units for k-1, Figure 11A) and maximum or PVHS (718 units for k-26, Figure 11S) values of hot maximum viscosity among the chufa flour samples was significant—421 units. The difference between the indicators of minimum viscosity in the hot state for the studied chufa flour samples decreased to 239 units; the minimum values (265 units) were noted for k-1 (Figure 11A), and the maximum (504 units) for k-14 (Figure 11J). The highest value of maximum viscosity in the cold state was noted for sample k-14 (Figure 11J), and the lowest (462 units) for k-1 (Figure 11A); the difference between the values was 734 units. At the end of the measurement, the minimum and maximum viscosity values or VEM were noted for samples k-14 (1108 units, Figure 11J) and k-1 (446 units, Figure 11A), with a slightly less significant difference of 662 units. The drop in viscosity at the maximum temperature, or VDPT, shows the resistance of the paste viscosity to physical load (mixing) at the maximum test temperature. Accordingly, the greater the drop, the lower the resistance [32]. This indicator in the samples ranged from 32 (k-1, Figure 11A) to 147 (k-13, Figure 11I), showing quite large variability. The rise in viscosity during cooling, or VIUC, and CPPV are associated with a change in the rheological properties of the water–flour mixture as the temperature decreases. The range of VIUC variability was significant. It ranged from 197 units (k-1, Figure 11A) to 692 units (k-14, Figure 11J), and the difference was 495 units. According to this indicator, the flour samples can be conditionally divided into a group with the lowest VIUC values (from 197 to 311 units), which included ten composite flour samples (k-1, 2, 7, 8, 9, 10, 11, 12, 24, 27, Figure 11A–H,R,T), with average values (from 364 units to 396 units) of four samples (k-16, 19, 20, 21, Figure 11L,N–P), and with high (from 434 units to 692 units) VIUC values of six composite flour samples (k-13, 14, 15, 17, 23, 26, Figure 11I–K,M,Q,S). The indicator of the rise in viscosity from the maximum values in the hot state to the maximum values in the cold state, or VIHC, shows the difference between the rise in viscosity during cooling and the drop in viscosity at the maximum temperature. The grouping of composite flour samples based on all three aforementioned indicators was identical. As can be seen from the above results, the range of variability among the samples for a number of visco-amylography indicators was quite high. Obviously, the result may be influenced by the amount of carbohydrates in the tubers of various chufa accessions and their quality. The difference between the minimum (26.08% for k-17) and maximum (45.69% for k-7) starch content was 19.61%. On average, for most samples, the starch content was 44.37%.

##### Farinographic Studies

WAC characterizes the technological properties of the raw material and determines how much water is needed to obtain a dough of standard consistency. Depending on the quality and grade of flour, the WAC can range from 40 to 75%. For the control sample, the WAC was 68%. In the composite samples (70% wheat flour and 30% chufa tuber flour), the WAC significantly decreased and averaged 53% (from 52% for k-14 to 55% for k-27). This is higher than the values obtained for bread with a 30% chufa content (49%) and corresponds to the WAC values for bread with a 20% chufa content (52%) [29]. DDT and DS characterize the strength of the flour. It is believed that the higher they are, the better the quality. In the composite flour samples, this indicator significantly increased compared to the control (3.68 and 1.3 min, respectively). The dough development time for the composite flour varied in the range from 6.17 min (k-26) to 18.48 min for flour with the addition of sample k-1. Dough stability ranged from 3.8 min (k-21) to 16.85 min (k-23). As a result, the dough in the mixed bread samples we studied became much stronger, which was significantly different from the values for bread with the same ratio of wheat flour and chufa tuber flour obtained by Shaban et al. (DDT-3.0 min and DS-9.5 min, respectively) [29]. DD shows how much the dough weakens during its prolonged mixing. The higher this indicator, the weaker the dough [33,34,35,36,37]. The control sample had a value of 80 units. In the composite samples, softening significantly decreased. The indicator value ranged from eight units (k-12) to thirty-four units (k-13). The VN indicator summarizes the assessment of the rheological properties of various flour samples obtained using the device. It is believed that the higher it is, the better the quality of the flour sample. The VN of the composite flour samples exceeded the control value of 51. For samples of wheat flour blended with flour from tubers of different chufa accessions, the VN ranged from 68 (k-26) to 94 (k-1), indicating that the quality of the composite flour in terms of rheological properties is much higher than that of the control wheat flour. Significant differences were noted for all studied indicators among the composite dough samples. As a result of the study, specific samples were highlighted: k-24 for WAC (57%), k-23 (15.03 min) and k-8 (16.9 min) for DDT; k-19 (16.8 min) and k-7 (16.03 min) for DS; and k-7 (91), k-2 (93), k-8 (93), and k-23 (91) for VN.

#### 3.3.3. Evaluation of Flour Color Indicators

The color of composite flour is determined by the presence of coloration from the endosperm and seed coats in the ground material of chufa tubers. The study of the color characteristics of flour from chufa tubers was carried out in the RGB color space, a color system that defines the final color as the result of adding three basic colors: red, green, and blue, within the device’s color model [38,39]. The least variable indicator for the flour samples was the brightness indicator *L* (*CV* less than 10%), from the minimum (43.92 for k-21) to the maximum (65.28 for k-19). For the main part of the samples, *L* was more than 50 units, on average, 59.5; i.e., the flour had different shades of beige. The indicators of the blue–yellow range *b,* the chroma index *C*, and the hue angle (*h*°) showed medium variability (*CV* from 10 to 20%). The values of the indicator b were positive with a maximum of 17.25 for k-11 and a minimum of 4.73 for k-21, with an average value of 15.3; i.e., all samples had a yellow color of varying intensity. The minimum chroma index *C* was 13.61 for k-1, the maximum was 28.49 for k-21, and the average was 16.6. The hue angle (*h*°) changed from 9.3° for k-21 to 88.07° for k-10, with an average of 81.8°. The main part of the flour samples had shades of yellow, while k-21 exhibited a red–purple hue. The most variable color indicator (*CV* more than 100%) for the studied flour samples was *a* (red–green range), from a minimum of 0.475 for k-10 to a maximum of 28.09 for k-21; the average was 2.6. The values of *a* for all samples were positive, i.e., they had shades of red of varying intensity. The color assessment of chufa tuber flour obtained by us is partially consistent with the data of Shaban et al. for bread with 30% chufa addition, where the *L* value was 58.95, *a* ws 2.15, but *b* was significantly higher (22.82). This is probably because Shaban et al. studied the color of baked bread from mixed flour, whereas in our case we provide values for chufa tuber flour, and also due to the difference in the material chosen for making bread [29].

#### 3.3.4. Baking Quality

Baking quality determines the commercial appeal of bakery products. SS corresponds to the ratio of the height (H) of the baked product to its diameter (D). SS less than 0.35 is considered weak; bread with such indicators has a flattened shape. Medium—from 0.35 to 0.45 and strong—more than 0.45. Bread with strong shape stability is characterized by an optimal ratio of the diameter and height of the baked product and is considered the most attractive to the consumer [39,40]. The first group consisted of three samples (k-15, 20, 26, Figure 12L,P,T), the second consisted of the majority of bread samples (k-2, 7, 10, 11, 12, 13, 14, 16, 17, 21, 23, 24, Figure 12C,D,G–K,M,N,Q–S) and the third consisted of k-8, 9, 19, 27 (Figure 12E,F,O–U). It should be noted that almost all values obtained for bread made from composite flour were higher than the shape stability of the control (Figure 12A), except samples k-15, 20 (Figure 12L,P). CD, g/mL, was determined as the ratio of the mass of a cylindrical crumb sample to the volume occupied by this cylinder. For almost all bread samples made from composite flour, except k-13 and k-26, the CD was higher than the control (0.53 g/mL) and ranged from 0.57 to 0.67 g/mL. However, the obtained results were lower than the CD values of bread made from wholegrain wheat flour (1.21 g/mL) [41,42]. CH (g load) for the control was 4901 g. For bread samples made from composite flour k-2, 10, 12, 13, 15, 16, 20, CH was lower (from 411 g for k-2 to 4686 g for k-15), while for the remaining bread samples, it was higher. S24 (g load) for the control was 2770 g. For bread samples with the addition of chufa k-1, 11, 12, 15, 16, 17, 21, 23, 24, the indicator was lower than the control (from 649 g for k-24 to 2663 g for k-12), while for the remaining samples it was higher. When comparing the values of the CH and S24 indicators, we can identify bread samples with the addition of chufa (k-12, 15, 16) in which both indicators were lower than the control, meaning that the staling rate was lower. TQS, bread with added chufa flour from various accessions was divided into three groups. The first group included products (k-17, 23, Figure 12N,R) with a score above four points, the second (k-1, 2, 7, 9, 10, 11, 14, 19, 20, 21, 24, 26, 27, Figure 12B–D,F–H,K,O–Q,T,U) was three points and above, and the third (k-8, 12, 13, 15, 16, Figure 12E,I,J,L,M) was below three points, which corresponded to good, satisfactory, and unsatisfactory taste.

As a result of the viscoamylographic analysis of the mixed dough, it was found that the maximum viscosity upon cooling was recorded for the sample k-14 (1108 units), which makes it possible to use it as a thickener in food production. Farinography showed that optimal parameters for high-quality mixed dough (DDT, VN, DS, DD) were recorded for dough samples with the addition of chufa k-1, k-23, k-12, and k-1, exhibiting values of 18.48 min, 94 units, 16.85 min, and 8 units, respectively. Good baking quality indicators such as shape stability were noted for the samples of mixed bread k-8, k- 9, k-19, k-27; CD—for k-13, k-26; CH and S24—for k-12, k-15, k-16; good TQS—for k-17, k-23. Thus, the flour of chufa accessions improves the mixed dough and bread quality. Hence, chufa tubers can be recommended for improving the baking properties of not only wheat but also other bready cereals, and for functional food production.

### 3.4. Results of Statistical Analysis of Technological Assessment

#### 3.4.1. Correlation Analysis of Technological Assessment Parameters 

The matrix of correlation coefficients and their significance are presented in Appendix A, respectively. A direct mutual influence (ρ = 0.000) was established between the baking evaluation indicators SS and CD (0.681); between the farinographic evaluation indicators (DDT) and the sedimentation analysis (PSC2) (0.191); and between the sedimentation analysis indicators PSC1 and PSC2 (0.798). Positive correlations were observed between the viscoamylography indicators PT and PTS; PVHS and MHPV; MHPV and CPPV; CPPV and VEM; TPV and PVT (0.952; 0.953; 0.959; 0.995; 0.742, respectively), while negative correlations were observed between TPV and PVHS; TPV and VDPT; PVT and VDPT (−0.699; −0.715; −0.809). An inverse relationship was established between the color indicators: brightness (*L1*, *L2*), yellow hue (*b1*, *b2*), hue angle (*h1*, *h2*) of the composite flour and the fiber content (−0.830; −0.824; −0.910; −0.910; −0.762; −0.765, respectively); a direct relationship was established between the indicators of red hue of the flour (*a1*, *a2*) and the fiber content (0.766; 0.765, respectively). With a reliability of 0.01 < ρ < 0.045, a direct influence was observed of oil content on the farinographic evaluation of dough DD (0.593); of AOA on baking qualities CD (0.589); of farinography indicators (DDT, DS, VN) on the amylography indicator TPV (0.500; 0.493; 0.529, respectively); of DD on VIUC (0.445); and of farinography indicators (DDT, DS, VN) on baking qualities (SS and CD) (r was approximately 0.5 or greater). The oil content negatively affected baking qualities SS (−0.473) and CD (−0.594); the fiber content affected sedimentation index PSC2 (−0.464), viscoamylographic PTS (−0.454), baking evaluation TQS (−0.492). With a reliability of 0.05 < ρ < 0.097, a direct relationship was established between the protein content and the farinography indicators DDT (0.382) and VN (0.391) between the PS and AOA content (0.390). Based on the results of the correlation analysis, the influence of biochemical composition characteristics (protein, oil, PS, AOA) on the technological evaluation data is evident. The taste qualities (TQS) reliably depended on the fiber content, sedimentation indicators (PSC1, PSC2), farinographic analysis (WAC), visco-amylography (MHPV), other baking qualities (S24, SS), and flour color indicators.

#### 3.4.2. Factor Analysis

Factor analysis of the system of correlations of traits of biochemistry and technological assessment allowed the isolation of eight factors, which collectively cover 90.7% of the total variability of the complex (Table 6). Factor 1 varies the content of oil, fiber, and the color traits of the chufa flour. Factor 2 varies mostly the traits of viscoamylograph assessment as follows: “maximum viscosity of starch in the hot state” (PVHS), “minimum viscosity in the hot state” (MHPV), “maximum viscosity in the cold state” (CPPV), “viscosity at the end of measurement” (VEM), “rise in viscosity during cooling” (VIUC), “rise in viscosity from maximum hot to maximum cold” (VIHC). Factor 3 varies fiber content in the flour samples, starch gelatinization onset temperature (PTS), chufa flour color traits, shape stability (SS). Factor 4 varies the starch content in the flour and bread staling after 24 h (S24). Factor 5 also varies the traits of viscoamylograph assessment as follows: starch gelatinization onset viscosity Time (IPV), maximum viscosity in the hot state (PVHS), maximum viscosity temperature (PVT), drop in viscosity at maximum temperature (VDPT). Factor 6 varies rheological properties of composite dough as follows: dough development time (DDT), dough stability (DS), and valorimeter number (VN). Factor 7 varies sedimentation analysis traits as follows: flour swelling capacity phase 1 (PSC1) and flour swelling capacity phase 2 (PSC2). Factor 8 varies antioxidant activity of the flour samples and the ratio phase 2/phase 1 (Phase 2/Phase 1).

The results of the analysis of variance showed that a significant influence of the origin of the chufa tubers accessions on the trait value was noted for the trait “gelatinization onset viscosity” (*p* = 0.005) and close to significant for three traits: “dough development time” (*p* = 0.107), “dough stability” (*p* = 0.101), and “valorimeter number” (VN) (*p* = 0.098) (Table 7).

#### 3.4.3. Discriminant Analysis by Origin

As a result of the discriminant analysis, classification functions were constructed (Table 8). Based on these functions, the chufa tuber accessions were differentiated by origin into three groups: Eastern Europe, Southern Europe, and Central Europe. These functions provide 100% correct classification (Table 9).

For two accessions, k-2 (Russia) *p* = 0.994 and k-16 (Africa) *p* = 0.985, the reliability of assignment to the observed group is below 1.00 (Table 10).

#### 3.4.4. Canonical Analysis

In the canonical analysis, two axes were identified—Root 1 (85.8% of variance), differentiating European and African accessions, and Root 2 (14.2%), differentiating accessions from Eastern, Central, and Southern Europe (Table 11).

#### 3.4.5. Discriminant Analysis

Discriminant analysis of chufa accessions of different origins based on baking qualities showed that k-2 stands out in the group from Eastern Europe, and k-16 in the group from Africa (Figure 13). K-2 differs from other accessions of Eastern European origin by its high VN (93) and IPV of more than 80 units. K-16 differs from other African accessions by its DS (9.18 min) and PTS (76.9 °C), and VN (79).

## 4. Conclusions

According to biochemical data, all samples of chufa tuber flour and bread with chufa flour addition surpassed the control, except for protein and starch content. The k-9 bread sample stood out with a high fiber content (5.16%), sum of phenolic substances (213.8 mg%), and AOA (24.2 mg%); k-10 with a high protein and oil content (17.94% and 9.42%, respectively); k-27 with oil and fiber content (9.42% and 4.40%, respectively); k-16 with a high fiber content (4.54%); k-24 with high sum of phenolic substances (212.1 mg%); k-19 and k-26 with high AOA (24.2 mg% and 24.4 mg%, respectively). As a result of the viscoamylographic analysis of the mixed dough, it was found that the maximum viscosity upon cooling was recorded for the sample k-14 (1108 units), which makes it possible to use it as a thickener in food production. Farinography estimation showed that the optimal parameters for high-quality mixed dough (DDT, VN, DS, DD) corresponded to the samples with the addition of chufa accessions k-1, k-23, k-12, and k-1 (18.48 min, 94 units, and 16.85 min, 8 units, respectively). Good baking quality indicators included SS for samples k-8, k-9, k-19, k-27; CD for k-13, k-26; CH and S24 for k-12, k-15, k-16; good TQS—for k-17, k-23. Thus, these chufa accessions improve the quality of mixed dough and bread and, therefore, can be used for the production of functional foods. Based on the results of the correlation analysis, the influence of biochemical composition characteristics (protein, oil, PS, AOA) on the technological evaluation data is evident. The taste qualities (TQS) reliably depended on the fiber content, sedimentation indicators (PSC1, PSC2), farinographic analysis (WAC), viscoamylography (MHPV), other baking qualities (S24, SS), and flour color indicators. According to the results of the farinographic study, it was found that the dough made from mixed flour was stronger than the control: it was more resistant to kneading and had a lower degree of liquefaction. Therefore, we can conclude that chufa tuber flour can improve the baking properties of whole-grain wheat flour. In terms of organoleptic properties, differences were also identified, and the samples that improved the taste of mixed bread were selected. Chufa samples with high rheological properties and high sedimentation values of the mixed dough can be recommended for improving the baking properties of wheat and other bread cereals. The presented results characterize the properties and potential of chufa tuber accessions grown in the conditions of the Kuban–Azov Plain as a material for regional functional food manufacturing and, in general, for the food industry.

In general, the data from the current study are preliminary. We have confirmed that chufa is a promising crop for bakery production in the region, given the unusual reproduction conditions for chufa. The next step will be the quality evaluation of mixed bread with different ratio of chufa flour addition, including chufa accessions grown in other regions of the Russian Federation (the Сentral Chernozem and Northwestern regions) as a part of the research program.

## Figures and Tables

**Figure 1 foods-14-03680-f001:**
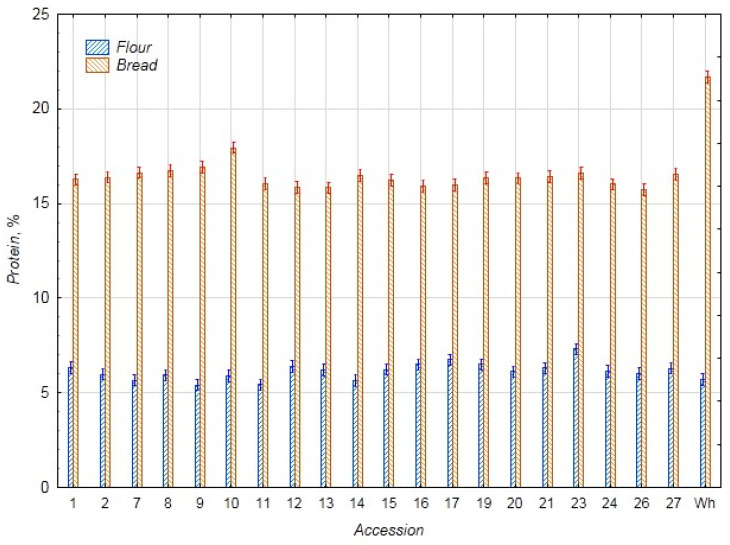
Protein content (%) in flour samples of chufa tubers, whole-grain wheat (Wh), composite bread, and whole-grain wheat bread (Wh).

**Figure 2 foods-14-03680-f002:**
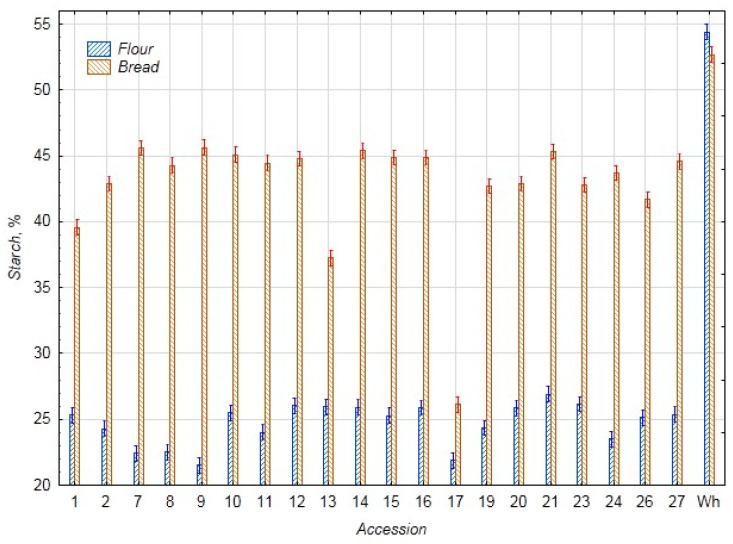
Starch content (%) in flour samples of chufa tubers, whole-grain wheat (Wh), composite bread, and whole-grain wheat bread (Wh).

**Figure 3 foods-14-03680-f003:**
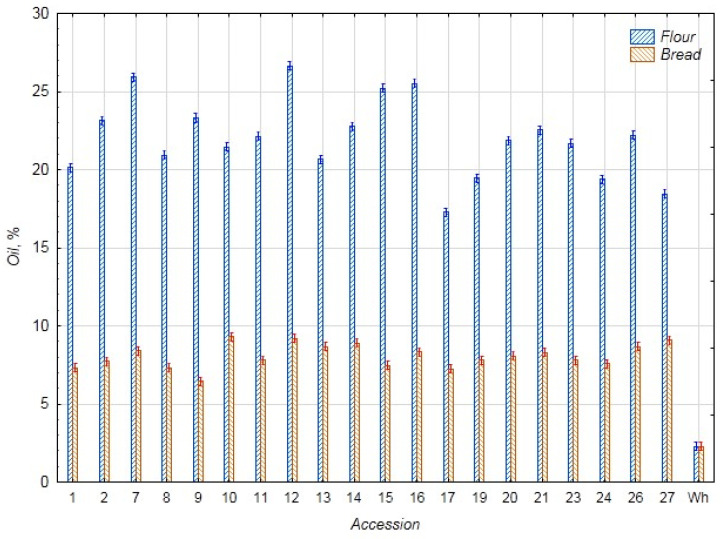
Oil content in flour samples of chufa tubers, whole-grain wheat (Wh), composite bread, and whole-grain wheat bread (Wh) (%).

**Figure 4 foods-14-03680-f004:**
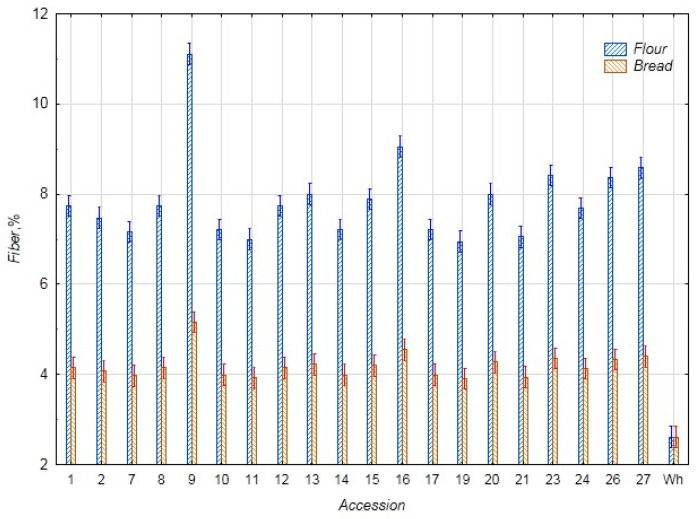
Fiber content in flour samples of chufa tubers, whole-grain wheat (Wh), composite bread, and whole-grain wheat bread (Wh) (%).

**Figure 5 foods-14-03680-f005:**
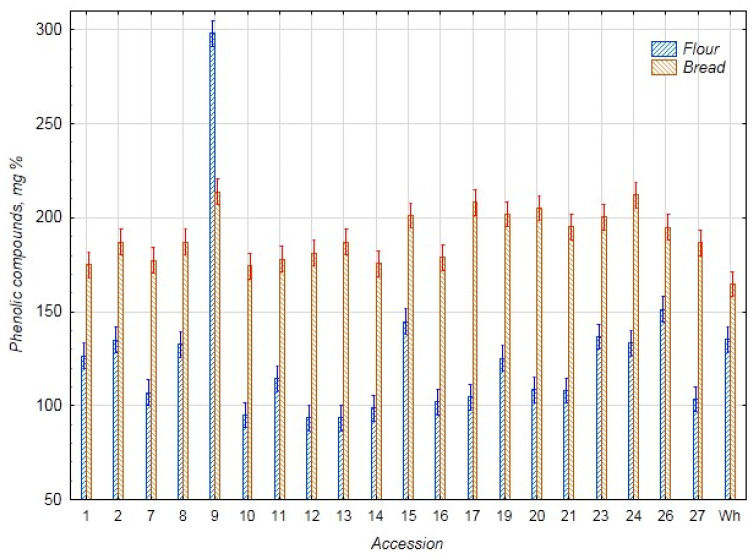
Content of total phenolic compounds (mg%) in flour samples of chufa tubers, whole-grain wheat (Wh), composite bread, and whole-grain wheat bread (Wh).

**Figure 6 foods-14-03680-f006:**
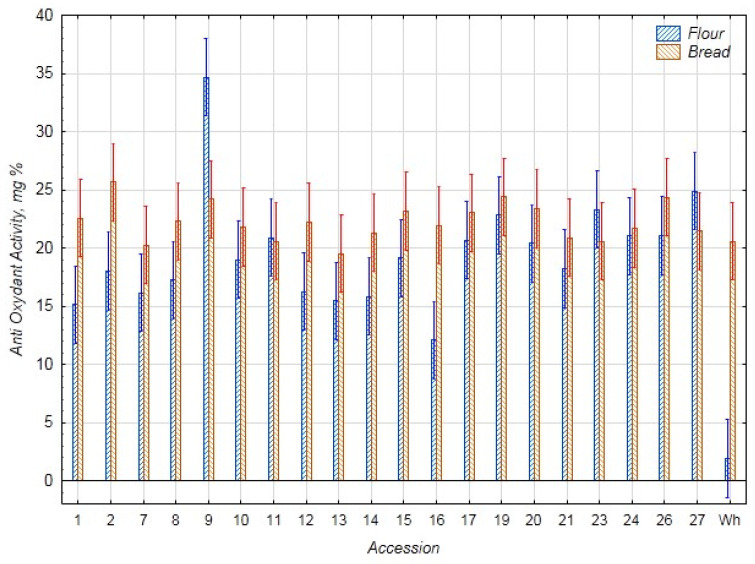
Antioxidant activity (AOA) in flour samples of chufa tubers and whole-grain wheat (Wh), composite bread, and wheat bread (Wh) (mg%).

**Figure 7 foods-14-03680-f007:**
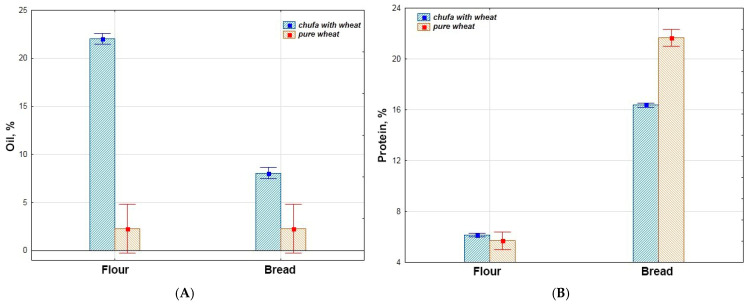
The share of influence of variation sources on the biochemical traits of chufa tuber flour, whole-grain wheat flour; and bread made from mixed flour and whole-grain wheat flour. (**A**)—oil, (**B**)—protein, (**C**)—starch, (**D**)—fiber, (**E**)—phenolic compounds, (**F**)—antioxidant activity.

**Figure 8 foods-14-03680-f008:**
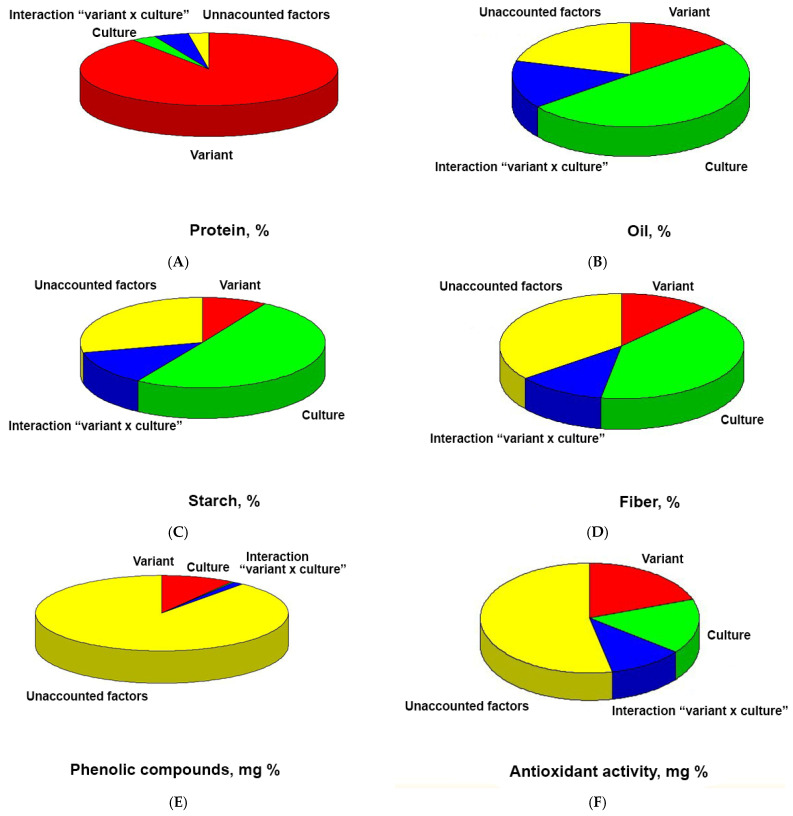
Sectoral diagrams showing the share of influence of factors: Variant—flour—bread; Culture—chufa, pure wheat; Variant*******Culture—different behavior of the trait under different combinations of variant and culture, on biochemical indicators. (**A**)—protein content, (**B**)—oil content, (**C**)—starch content, (**D**)—fiber content, (**E**)—phenolic compounds content, (**F**)—OAO of chufa tuber flour and composite bread.

**Figure 9 foods-14-03680-f009:**
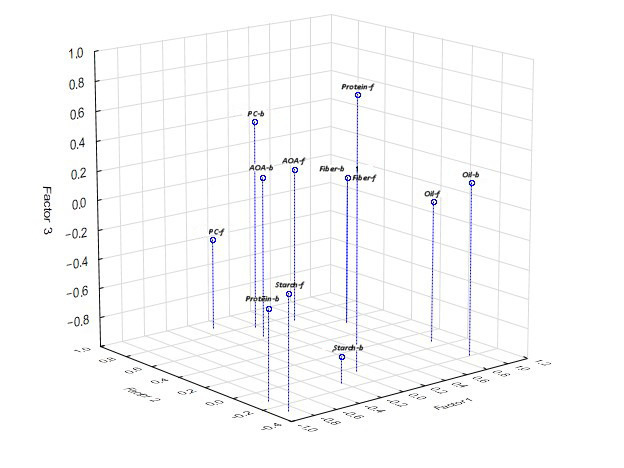
Factor structure of the variation of biochemical traits of chufa and whole-grain wheat flour (f) and bread samples (b).

**Figure 10 foods-14-03680-f010:**
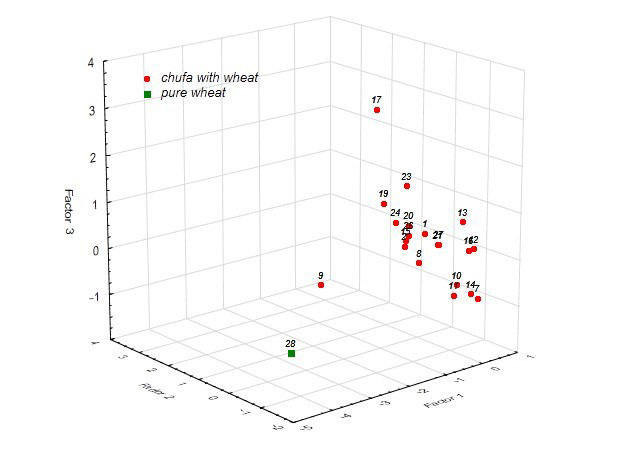
Differentiation of chufa and whole-grain wheat samples in the factor space.

**Figure 11 foods-14-03680-f011:**
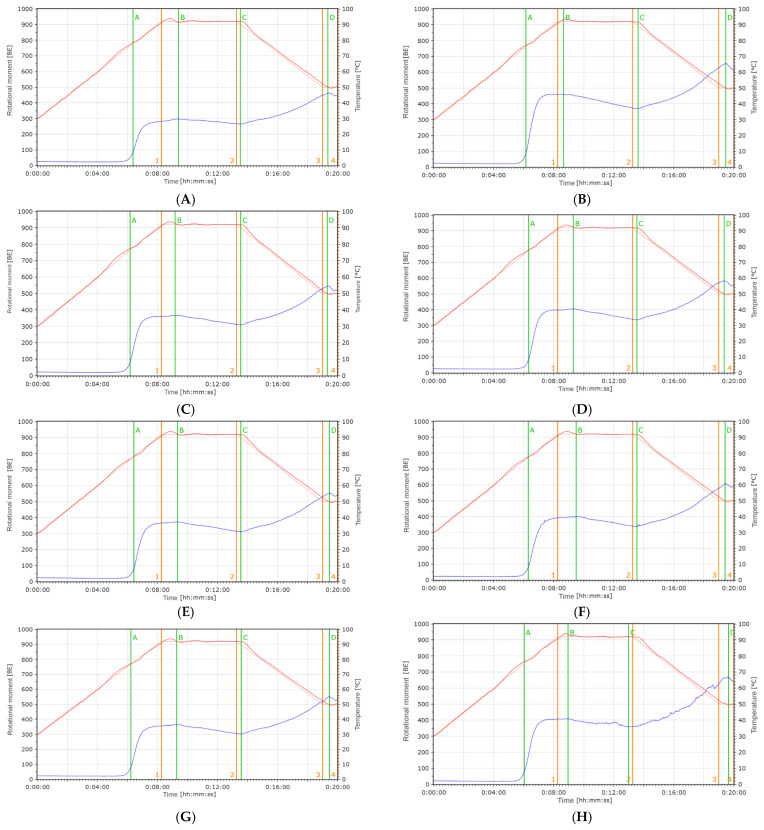
Viscoamylograms of flour samples with chufa addition (blue line indicates rotational moment; orange line—number of revolutions; red line—temperature; numbers 1, 2, 3, 4—phases of thermodynamic tests; letters **А, B, С, D**—indicators of thermodynamic tests). The green line indicates the main phases of the sample's rheological transformations: A—onset of gelatinization; B—maximum viscosity in the hot state; C—minimum viscosity in the hot state; D—maximum viscosity upon cooling.

**Figure 12 foods-14-03680-f012:**
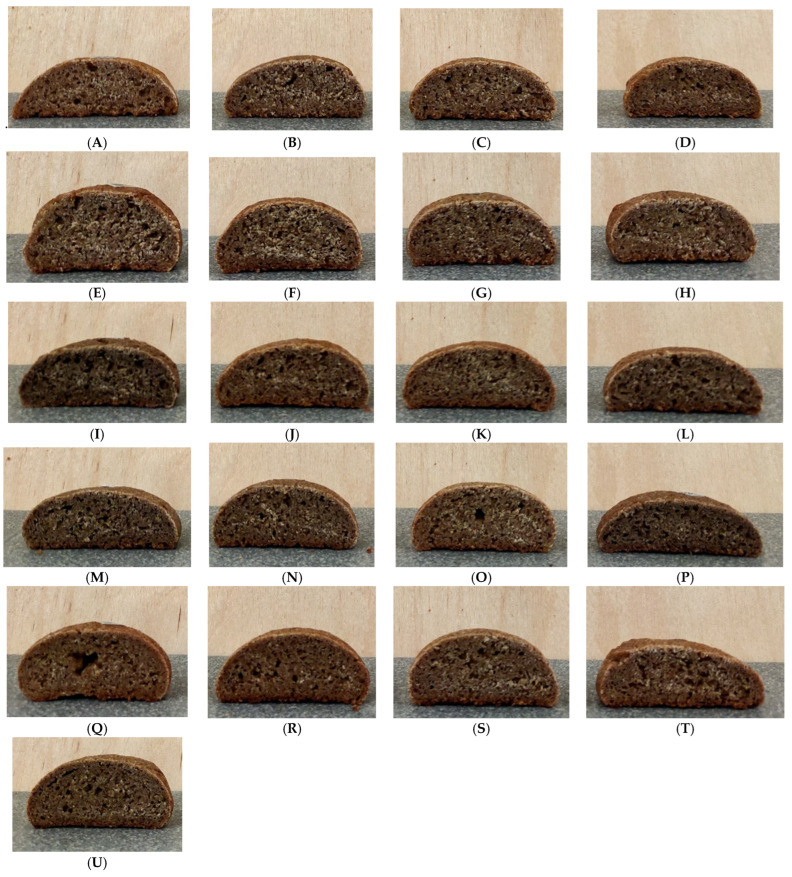
External view of bread from 100% whole-grain wheat flour as control, (**A**) bread from composite of 70% whole-grain wheat flour and 30% from chufa tubers accessions. (**B**)—k-1, (**C**)—k-2, (**D**)—k-7, (**E**)—k-8, (**F**)—k-9, (**G**)—k-11, (**H**)—k-10, (**I**)—k-12, (**J**)—k-13, (**K**)—k-14, (**L**)—k-15, (**M**)—k-16, (**N**)—k-17, (**O**)—k-19, (**P**)—k-20, (**Q**)—k-21, (**R**)—k-23, (**S**)—k-24, (**T**)—k-26, (**U**)—k-27.

**Figure 13 foods-14-03680-f013:**
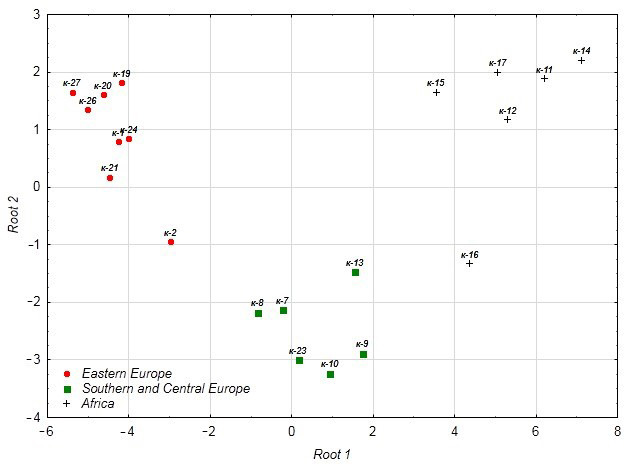
Discrimination of chufa tuber accessions of different origins by technological qualities.

**Table 1 foods-14-03680-t001:** List of *C. esculentus* accessions from the VIR collection used as a research material.

Сatalog Number	Origin
1	Russia
2	Russia
7	Poland
8	Bulgaria
9	Bulgaria
10	Bulgaria
11	Mali
12	Benin
13	Germany
14	Ivory Coast
15	Ivory Coast
16	Ivory Coast
17	Ivory Coast
19	Russia
20	Ukraine
21	Belarus
23	France
24	Ukraine
26	Russia
27	Russia

**Table 2 foods-14-03680-t002:** Biochemical parameters of chufa tuber flour, whole-grain wheat flour samples, and bread from mixed flour and whole-grain wheat flour.

Level	Flour				Bread			
	Chufa		Wheat		Chufa 30%/Wheat 70%		Wheat 100%	
	X	Sx	X	Sx	X	Sx	X	Sx
Oil, %	22.0	0.39	2.3	0.02	8.1	0.12	2.3	0.00
Protein, %	6.2	0.07	5.7	0.00	16.4	0.08	21.7	0.08
Starch, %	24.7	0.25	54.4	0.57	42.7	0.70	52.7	0.00
Fiber, %	7.9	0.15	2.6	0.00	4.2	0.05	2.6	0.00
Phenolic substances, mg%	125.6	6.96	135.2	0.35	191.0	2.07	164.8	2.95
Antioxidant activity, mg%	19.6	0.78	1.9	0.09	22.3	0.39	20.6	0.35

X—Average value of biochemical parameters. Sx—standard deviation

**Table 3 foods-14-03680-t003:** Correlation map of the biochemical traits of chufa tuber flour, composite bread (30% chufa/70% wheat), and their significance levels.

	Oil	Protein	Starch	Fiber	PC	АОА	B/Oil	B/Pr	B/St	B/F	B/PC	B/АОА
Oil	1.000	0.068	−0.826	0.728	−0.001	0.453	0.800	−0.811	−0.136	0.728	0.213	0.189
Protein	0.771	1.000	−0.134	0.124	−0.313	0.036	0.201	−0.295	−0.459	0.125	0.312	−0.004
Starch	0.000	0.563	1.000	−0.784	−0.069	−0.698	−0.766	0.887	0.490	−0.783	−0.469	−0.285
Fiber	0.000	0.593	0.000	1.000	0.443	0.775	0.568	−0.708	−0.252	1.000	0.524	0.315
PC	0.995	0.167	0.767	0.044	1.000	0.564	−0.379	0.112	0.172	0.439	0.468	0.429
АОА	0.039	0.878	0.000	0.000	0.008	1.000	0.384	−0.509	−0.249	0.774	0.722	0.389
B/Oil	0.000	0.383	0.000	0.007	0.090	0.086	1.000	−0.784	−0.277	0.568	0.087	0.014
B/Pr	0.000	0.194	0.000	0.000	0.628	0.019	0.000	1.000	0.534	−0.708	−0.433	−0.224
B/St	0.557	0.037	0.024	0.269	0.456	0.277	0.225	0.013	1.000	−0.253	−0.404	−0.177
B/F	0.000	0.589	0.000	0.000	0.047	0.000	0.007	0.000	0.269	1.000	0.527	0.318
B/PC	0.353	0.169	0.032	0.015	0.032	0.000	0.708	0.050	0.069	0.014	1.000	0.435
B/АОА	0.412	0.987	0.210	0.165	0.052	0.081	0.952	0.329	0.442	0.160	0.049	1.000

PC—phenolic compounds, АОА—antioxidant activity, B/Oil—oil content of composite bread, B/Pr—protein content of composite bread, B/St—starch content of composite bread, B/F—fiber content of composite bread, B/PC—phenolic compounds content of composite bread, B/АОА—antioxidant activity of composite bread. Correlation color map: r >= scarlet: −1.00–−0.80; red: −0.80–−0.20; pink: −0.20–−0.00; grey: 0.00–−0.20; light blue: 0.20–0.40; blue: 0.40–1.00.. *p*-value color map for correlations: *p* <= scarlet: 0.000–0.001; red: 0.001–0.010; light red: 0.010–0.050; pink: 0.050–0.150; light blue: 0.150–0.500; blue: 0.500–1.000.

**Table 4 foods-14-03680-t004:** Reliability and proportion of the source influence of the biochemical traits of chufa tubers and whole-grain wheat flour samples, composite bread (30% chufa/70% wheat) and whole-grain wheat bread samples, and their significance levels.

Sign	Reliability (*p*)			(*η*^2^) The Proportion of The Source’s Influence			
	Variant	Culture	Variant*Culture	Variant	Culture	Variant*Culture	У
Oil, %	0.000	0.000	0.000	14.8	49.6	14.8	20.8
Protein, %	0.000	0.000	0.000	90.0	3.1	4.4	2.6
Starch, %	0.000	0.000	0.000	8.5	50.4	12.5	28.7
Fiber, %	0.000	0.000	0.000	11.8	40.9	11.8	35.5
Phenolic substances, mg%	0.005	0.613	0.279	9.3	0.3	1.3	89.1
Antioxidant activity, mg%	0.000	0.000	0.000	19.5	16.2	11.0	53.4

Variant—flour, bread. Culture—chufa, pure wheat. Variant*Culture—different behavior of the trait under different combinations of variant and culture. U—unaccounted factors.

**Table 5 foods-14-03680-t005:** Factor structure of the variation of biochemical traits of chufa tuber flour and whole-grain wheat flour samples, composite bread (30% chufa/70% wheat), and whole-grain wheat bread samples.

	Factor 1	Factor 2	Factor 3
Oil	0.937	0.068	−0.034
Protein	0.051	−0.044	0.828
Starch	−0.878	−0.286	−0.237
Fiber	0.768	0.555	0.022
PC	−0.098	0.885	−0.375
АОА	0.510	0.746	0.061
B/Oil	0.913	−0.206	0.176
B/Pr	−0.856	−0.131	−0.387
B/St	−0.215	−0.140	−0.815
B/F	0.768	0.555	0.024
B/PC	0.169	0.774	0.432
B/АОА	0.070	0.640	0.101
The variance proportion by factor	39.6	26.1	16.1

**Table 6 foods-14-03680-t006:** The factor structure of variation in technological assessment traits of chufa tubers and whole-grain wheat dough samples, composite bread (30% chufa/70% wheat) and whole-grain wheat bread samples.

	Factor 1	Factor 2	Factor 3	Factor 4	Factor 5	Factor 6	Factor 7	Factor 8
Oil	0.648	−0.121	−0.263	−0.407	−0.357	−0.130	−0.252	−0.007
Protein	−0.403	0.261	0.523	−0.349	−0.290	0.259	−0.098	0.125
Starch	−0.101	−0.088	−0.199	−0.621	−0.071	0.212	−0.048	0.028
Fiber	−0.628	0.153	−0.652	0.054	0.124	0.013	−0.110	−0.081
PC	−0.454	0.215	−0.140	0.509	−0.028	−0.003	0.549	−0.177
АОА	−0.275	0.215	0.113	0.257	0.080	0.258	0.025	−0.718
WAC	0.191	0.569	−0.359	−0.216	−0.150	−0.135	0.593	−0.132
DDT	0.017	0.307	0.210	−0.051	0.169	0.889	0.005	0.009
DS	−0.169	0.319	−0.013	−0.070	0.258	0.809	0.079	−0.120
DD	0.361	−0.362	−0.010	−0.363	−0.413	0.049	0.073	0.217
VN	−0.105	0.375	0.116	−0.037	0.213	0.871	−0.000	−0.020
PSC1	0.263	−0.054	0.233	0.038	0.033	0.042	0.896	−0.116
PSC2	0.221	0.140	0.482	0.190	0.097	0.063	0.684	0.331
P2-1	0.041	0.323	0.496	0.255	0.124	0.054	−0.059	0.696
PT	0.110	−0.507	0.549	0.043	0.570	0.112	0.193	−0.051
IPV	0.363	0.174	−0.033	0.257	−0.628	−0.030	0.205	−0.340
PTS	0.010	−0.464	0.684	0.051	0.503	0.075	0.150	−0.107
TPV	−0.055	0.223	0.155	−0.200	0.799	0.288	0.141	0.197
PVHS	0.050	−0.764	0.011	0.107	−0.532	−0.208	−0.157	0.101
PVT	−0.098	0.208	−0.178	0.131	0.886	0.135	0.027	0.031
MHPV	0.034	−0.882	−0.071	0.093	−0.307	−0.196	−0.211	0.096
CPPV	0.056	−0.959	−0.068	−0.001	−0.176	−0.188	−0.057	0.009
VEM	0.080	−0.934	−0.114	0.001	−0.236	−0.194	−0.095	0.004
VDPT	0.073	−0.328	0.084	0.083	−0.833	−0.101	0.010	0.169
VIUC	0.066	−0.967	−0.063	−0.055	−0.093	−0.176	0.034	−0.042
VIHC	0.050	−0.940	−0.125	−0.095	0.176	−0.130	0.044	−0.076
*L1*	0.421	0.191	0.849	0.069	−0.094	0.077	0.097	0.079
*L2*	0.410	0.166	0.857	0.063	−0.083	0.103	0.046	0.092
*a1*	−0.959	0.036	−0.241	−0.009	0.052	0.036	−0.090	−0.037
*a2*	−0.958	0.034	−0.240	−0.014	0.054	0.041	−0.096	−0.033
*b1*	0.716	−0.007	0.678	0.029	−0.057	0.027	0.104	0.034
*b2*	0.720	−0.011	0.670	0.040	−0.041	0.008	0.121	0.046
*С1*	−0.970	0.059	0.106	0.001	0.056	0.079	−0.077	−0.056
*С2*	−0.975	0.047	0.081	0.018	0.066	0.074	−0.055	−0.050
*h1*	0.959	−0.036	0.240	0.012	−0.049	−0.039	0.091	0.027
*h2*	0.959	−0.033	0.241	0.011	−0.048	−0.037	0.093	0.031
SS	−0.003	0.407	0.713	0.121	0.257	0.232	0.274	−0.169
CD	−0.157	0.244	0.462	0.361	0.563	0.323	−0.106	−0.255
CH	−0.002	−0.294	0.130	0.523	−0.229	0.546	0.308	−0.216
S24	0.053	0.266	0.104	−0.703	0.406	−0.133	−0.058	−0.095
TQS	0.428	0.284	0.219	0.190	0.127	0.242	0.552	0.064
share of total	21.8	17.9	14.3	5.9	11.8	8.1	6.8	4.1

**Table 7 foods-14-03680-t007:** Reliability of the influence of chufa tuber accession origin on the technological assessment traits of bread (30% chufa/70% wheat) samples.

Trait	Abbreviation	*p*
Oil	Oil	0.975
Protein	Protein	0.098
Starch	Starch	0.817
Fiber	Fiber	0.510
Phenolic compounds	PC	0.576
Antioxidant activity	АОА	0.162
Water absorption capacity	WAC	0.250
Dough development time	DDT	0.107
Dough stability	DS	0.101
Dilution of the dough	DD	0.612
Valorimeter number	VN	0.098
Flour swelling capacity phase 1	PSC1	0.148
Flour swelling capacity phase 2	PSC2	0.541
Phase 2/Phase 1	P2-1	0.778
Pasting temperature	PT	0.350
Initial pasting viscosity	IPV	0.005
Pasting temperature start	PTS	0.188
Time to peak viscosity	TPV	0.934
Peak viscosity in the hot state	PVHS	0.791
Peak viscosity temperature	PVT	0.380
Minimum hot paste viscosity	MHPV	0.554
Cold paste peak viscosity	CPPV	0.433
Viscosity at the end of measurement	VEM	0.500
Viscosity drop at peak temperature	VDPT	0.528
Viscosity increase upon cooling	VIUC	0.388
Viscosity increase from hot to cold	VIHC	0.269
*L1* Flour color brightness	*L1*	0.832
*L2* Flour color brightness	*L2*	0.925
*a1* The red–green range of the spectrum	*a1*	0.300
*a2* The red–green range of the spectrum	*a2*	0.282
*b1* Yellow–blue range of the spectrum	*b1*	0.393
*b2* Yellow–blue range of the spectrum	*b2*	0.349
*С1* Chromaticity index	*c1*	0.416
*С2* Chromaticity index	*c2*	0.437
*h1* Shade angle	*h1*	0.272
*h2* Shade angle	*h2*	0.280
Shape stability	SS	0.824
Crumb density	CD	0.807
Crumb hardness	CH	0.186
Staling in 24 h	S24	0.421
Taste quality score	TQS	0.114

**Table 8 foods-14-03680-t008:** Classification functions for groups of chufa accessions of different origin from the VIR collection based on biochemical and technological assessment indicators.

	Eastern Europe	South and Central Europe	Africa
Initial pasting viscosity (IPV)	−166.4	−172.1	−174.2
Dough stability (DS)	−445.3	−452.3	−458.2
Peak viscosity temperature (PVT)	260.6	259.5	263.9
Pasting temperature Start (PTS)	2718.7	2740.6	2775.1
Pasting temperature (PT)	−11,199.4	−11,304.5	−11,443.1
Valorimeter number (VN)	450.0	457.9	463.1
Oil	1022.1	1034.7	1047.6
Dough development time (DDT)	−750.5	−762.2	−771.1
Constant number	−90,778.1	−91,787.8	−94,197.1

**Table 9 foods-14-03680-t009:** The results of the classification analysis based on different geographical origins of chufa tuber accessions.

	Correct Decisions	Eastern Europe	South and Central Europe	Africa
Eastern Europe	100.0	8	0	0
South and Central Europe	100.0	0	6	0
Africa	100.0	0	0	6
Total	100.0	8	6	6

**Table 10 foods-14-03680-t010:** Posterior probabilities of chufa tuber accessions belonging to groups of different geographical origins.

Catalog	The Observed Classification	Russia	Europe	Africa
k-1	Russia	1.000	0.000	0.000
k-2	Russia	0.994	0.006	0.000
k-7	Europe	0.000	1.000	0.000
k-8	Europe	0.000	1.000	0.000
k-9	Europe	0.000	1.000	0.000
k-10	Europe	0.000	1.000	0.000
k-11	Africa	0.000	0.000	1.000
k-12	Africa	0.000	0.000	1.000
k-13	Europe	0.000	1.000	0.000
k-14	Africa	0.000	0.000	1.000
k-15	Africa	0.000	0.000	1.000
k-16	Africa	0.000	0.015	0.985
k-17	Africa	0.000	0.000	1.000
k-19	Russia	1.000	0.000	0.000
k-20	Russia	1.000	0.000	0.000
k-21	Russia	1.000	0.000	0.000
k-23	Europe	0.000	1.000	0.000
k-24	Russia	1.000	0.000	0.000
k-26	Russia	1.000	0.000	0.000
k-27	Russia	1.000	0.000	0.000

**Table 11 foods-14-03680-t011:** Standardized coefficients for canonical variables based on the results of biochemical and technological assessment.

Sign	Root 1	Root 2
Initial pasting viscosity (IPV)	−2.115	1.221
Dough stability (DS)	−5.110	0.494
Peak viscosity temperature (PVT)	0.635	1.600
Pasting temperature start (PTS)	5.423	1.814
Pasting temperature (PT)	−4.474	−0.961
Valorimeter number (VN)	10.659	−2.759
Oil	2.146	0.103
Dough development time (DDT)	−7.375	1.127
Variance Explained by the Factor	85.8	14.2

## Data Availability

The original contributions presented in this study are included in the article/Appendix A. Further inquiries can be directed to the corresponding author.

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
