# Peer review of "Technological Assessment of Bread with the Addition of *Cyperus esculentus* L. Accessions Flour Grown in the Kuban–Azov Plain"

_foods, 2025, doi:10.3390/foods14213680_

Round 1
Reviewer 1 Report
Comments and Suggestions for Authors
This study evaluated 20 accessions of chufa from the VIR collection in Russia and their use in composite bread. A systematic evaluation was conducted covering biochemical composition (protein, starch, oil, fiber, phenolic compounds, and antioxidant activity), dough rheological properties, and baking quality traits. Overall, the article presents detailed data and makes a useful contribution by summarizing and synthesizing findings, offering guidance for future research in this field. Nevertheless, several aspects still require improvement.
- Although the introduction states that composite bread was prepared with 30% chufa and 70% wheat, the rationale behind choosing this exact ratio is not justified. Please clarify whether this proportion was selected based on literature, preliminary experiments, or technological considerations.
- The introduction should be improved by explaining why the Krasnodar region was chosen for this study and by more clearly summarizing the main innovative aspects of the work.
- The Methods section is somewhat difficult to follow. It would benefit from adding clearer subheadings or numbering (e.g., 2.1 Plant material, 2.2 Biochemical analysis, 2.3 Rheological tests, 2.4 Baking evaluation, 2.5 Statistical analysis) to improve readability and logical flow.
- In Line 84: The biochemical analyses were performed with only two biological replicates. Please justify why only two replicates were used, and discuss whether additional replication would strengthen the reliability of the results.
- The description of the rheological tests lacks detail on processing and sample conditions. Please clarify the test settings (e.g., temperature, mixing time) and sample preparation to ensure reproducibility.
- Two sections are both labeled as “” (“Results and discussion”and “Baking qualities”). Please revise the section numbering to ensure clarity and consistency.
- It is suggested to revise the paragraph structure. Currently, many paragraphs consist of only one or two sentences, which makes the text appear fragmented. Merging related content into fuller paragraphs would improve readability and flow.
- The results are mostly descriptive and lack deeper analysis.It is recommended to expand the discussion by linking physicochemical properties with their mechanisms in dough formation and bread quality, and by comparing regional accessions to highlight targeted applications.
- The Discussion relies mainly on the study’s own data, without sufficient comparison to previous research. In addition, the reasons for discrepancies with prior studies are not analyzed. A deeper literature-based comparison would strengthen the discussion and highlight the study’s contribution.
- The conclusion mainly summarizes average biochemical values and selected accessions,but does not explain the specific advantages of the highlighted accessions for regional application. In addition, the study’s limitations are not discussed. It is recommended to strengthen the conclusion by linking results to the research aim, clarifying practical implications, and adding limitations and future perspectives.
Author Response
Answer to Reviewer 1 Comments
Comments and Suggestions for Authors
This study evaluated 20 accessions of chufa from the VIR collection in Russia and their use in composite bread. A systematic evaluation was conducted covering biochemical composition (protein, starch, oil, fiber, phenolic compounds, and antioxidant activity), dough rheological properties, and baking quality traits. Overall, the article presents detailed data and makes a useful contribution by summarizing and synthesizing findings, offering guidance for future research in this field. Nevertheless, several aspects still require improvement.
The authors express their deep gratitude to the reviewer for his time and comments.
Comment
Although the introduction states that composite bread was prepared with 30% chufa and 70% wheat, the rationale behind choosing this exact ratio is not justified. Please clarify whether this proportion was selected based on literature, preliminary experiments, or technological considerations.
Answer
Our task was to evaluate the potential of different chufa tuber accessions from the VIR collection for bakery production.
The evaluation of the baking properties of raw materials often begins with investigations of the rheological properties of dough. Similarly, in our research, we tested the rheological properties of a composite dough in comparison with a control (dough made from wheat flour of the spring soft wheat variety “Pamyati Suslyakova”) to understand the nature and direction of the interaction between the chufa tuber flour and wheat whole grain flour during technological processing. Another primary goal of the research was to determine whether there are differences between the various chufa accessions in their baking application. These differences were recorded during the preliminary stage of studying the composite dough's rheology. It was also noted that these differences are most obvious at a 70:30 wheat-to-chufa ratio. That is, the optimal ratio of chufa tuber flour to whole grain wheat flour was selected. Based on our research, we have identified several samples of chufa tubers with optimal technological qualities, biochemical composition, and taste properties for further study.
In the future, we plan to study bakery products made from composite flour with different ratios of chufa and wheat flour (40/60, 50/50, and 70/30%), as well as marzipan-type products made solely from chufa tuber flour.
Comment
The introduction should be improved by explaining why the Krasnodar region was chosen for this study and by more clearly summarizing the main innovative aspects of the work.
Answer
Currently, the industrial-scale cultivation of chufa exists only in Spain and Italy, i.e., in Southern European countries. Within the Russian Federation, one of the most suitable regions in terms of climatic conditions for cultivating chufa is the Krasnodar region. Furthermore, chufa is grown, though not on an industrial scale, in the Tambov region, Astrakhan region, and the Leningrad region (VIR experimental fields). Although chufa is not a crop typical of the Russian Federation, breeding work is being carried out based on the VIR collection, which has resulted in the development of the chufa variety "VIR 1," currently in the approval stage. It is important to note that the study of chufa is generating significant interest among agricultural producers in the Krasnodar region.
Thus, the aim of our work was to evaluate the potential of the VIR chufa collection for regional bakery production (including functional food products) and the regional food industry as a whole.
Comment
The Methods section is somewhat difficult to follow. It would benefit from adding clearer subheadings or numbering (e.g., 2.1 Plant material, 2.2 Biochemical analysis, 2.3 Rheological tests, 2.4 Baking evaluation, 2.5 Statistical analysis) to improve readability and logical flow.
Answer
The Materials and Methods section has been revised
Comment
In Line 84: The biochemical analyses were performed with only two biological replicates. Please justify why only two replicates were used, and discuss whether additional replication would strengthen the reliability of the results.
Answer
You are absolutely right; the more replicates, the more reliable the result. However, our adjustments are due to the necessity of analyzing a large number of samples simultaneously. The number of biological replicates for each analysis was chosen by adapting the analytical procedure for the screening of a large sample set. The data obtained from the study of biological replicates are used for subsequent statistical analysis provided that the variation between replicates is within the instrument's technical error margin. Otherwise, additional biological replicates are analyzed until the required consistency is achieved. It should be noted that the regional regulatory documents recommend a minimum of two biological replicates.
Comment
The description of the rheological tests lacks detail on processing and sample conditions. Please clarify the test settings (e.g., temperature, mixing time) and sample preparation to ensure reproducibility.
Answer
The text has been revised according to the comments.
Comment
Two sections are both labeled as “” (“Results and discussion”and “Baking qualities”). Please revise the section numbering to ensure clarity and consistency.
Answer
The text has been revised according to the comments.
Comment
It is suggested to revise the paragraph structure. Currently, many paragraphs consist of only one or two sentences, which makes the text appear fragmented. Merging related content into fuller paragraphs would improve readability and flow.
Answer
The text has been revised according to the comments.
Comment
The results are mostly descriptive and lack deeper analysis. It is recommended to expand the discussion by linking physicochemical properties with their mechanisms in dough formation and bread quality, and by comparing regional accessions to highlight targeted applications.
Answer
We have expanded the discussion of the results, comparing the current findings with our previous results (2020-21) and with data from other researchers. Unfortunately, however, we must note that there are few such studies.
We have also provided a summary of the results for each section (biochemical and technological assessment) and highlighted the chufa samples that show potential for various branches of the food industry.
The correlation results of biochemical and technological assessments of flour and bread from different chufa tuber accessions are presented in the statistical processing sections.
In a previous study, we assessed the biochemical properties of chufa grown in the Krasnodar and Tambov regions. A comparison of our previous data and current values is presented.
The data from the current study are preliminary. Our primary goal was to evaluate the potential of chufa for bakery production. The next stage of research will involve assessing the baking and biochemical qualities of composite bread with different ratios of chufa tuber flour to wheat flour (40:60, 50:50, and 70:30), as well as studying flour from chufa tubers cultivated in other regions: the Central Black Earth and the North-Western regions of Russia.
Comment
The Discussion relies mainly on the study’s own data, without sufficient comparison to previous research. In addition, the reasons for discrepancies with prior studies are not analyzed. A deeper literature-based comparison would strengthen the discussion and highlight the study’s contribution.
Answer
We have expanded the discussion of the results. The current findings were compared with our previous results (2020-21) and with data from other researchers. Conclusions have been drawn regarding the reasons for the discrepancies between the results of other researchers and our previously obtained data.
Comment
The conclusion mainly summarizes average biochemical values and selected accessions, but does not explain the specific advantages of the highlighted accessions for regional application. In addition, the study’s limitations are not discussed. It is recommended to strengthen the conclusion by linking results to the research aim, clarifying practical implications, and adding limitations and future perspectives.
Answer
The Conclusion has been revised in accordance with the recommendations.
All corrections in the text are marked in green.

Reviewer 2 Report
Comments and Suggestions for Authors
The manuscript by Kon’kova et al. reports on the biochemical composition and technological evaluation of chufa tubers and their utilization in composite bread formulations. The topic is relevant, the experimental design is appropriate, and the data provide valuable insights into the use of Cyperus esculentus for functional bakery applications. The paper fits well within the journal’s scope and can make a useful contribution to the field of food chemistry and technology. However, several issues need to be addressed and clarified before it can be considered for publication.
Major Comments:
- The title of the manuscript should be revised. The acronym VIR is unclear to international readers. Please either expand it (Vavilov Institute of Plant Genetic Resources) or consider simplifying the title.
- The abstract should be rewritten to include more quantitative data. Clearly state the objective, methods, main findings, and practical significance in a concise way. Please avoid overly general sentences and emphasize the main outcomes and how they advance knowledge in functional bakery formulations.
- The discussion section would benefit from a stronger comparison with previous studies, especially regarding biochemical composition and baking performance. Include references to similar studies on composite flour breads. This will help contextualize your findings and highlight the novelty of your work.
- While descriptive statistics, ANOVA, and factor analysis are presented, it would help to clarify the number of replicates for each test and the confidence levels. Consider adding letters or symbols on graphs to indicate significant differences where applicable.
- Figures 1 (A, B), 2 (A, B), 3 (A, B), and 4 (A, B) should be merged into single composite figures with clear sub-labels (A–B). This will improve readability and reduce redundancy.
- Although the study is comprehensive, the novelty statement should be emphasized in both the Introduction and Conclusion. Explicitly state how these findings can contribute to developing functional bread products or valorizing chufa tubers in the food industry.
- Please expand the Conclusion slightly to summarize main findings quantitatively, note key chufa accessions with superior performance, and suggest potential industrial applications.
- Some minor editing in English is required. Several sentences are long and could be simplified to improve clarity and flow.
Minor Comments
- Use periods, not commas, e.g. 6.15%, not 6,15%
- Define all abbreviations at first use
- Ensure unit consistency throughout the manuscript
- Consider adding “functional foods,” “composite flour,” and/or “technological evaluation” as keywords
Comments on the Quality of English LanguageSome minor editing in English is required. Several sentences are long and could be simplified to improve clarity and flow.
Author Response
Answer to Reviewer 2 Comments
Comments and Suggestions for Authors
The manuscript by Kon’kova et al. reports on the biochemical composition and technological evaluation of chufa tubers and their utilization in composite bread formulations. The topic is relevant, the experimental design is appropriate, and the data provide valuable insights into the use of Cyperus esculentus for functional bakery applications. The paper fits well within the journal’s scope and can make a useful contribution to the field of food chemistry and technology. However, several issues need to be addressed and clarified before it can be considered for publication.
The authors express their deep gratitude to the reviewer for his time and comments.
Major Comments:
The title of the manuscript should be revised. The acronym VIR is unclear to international readers. Please either expand it (Vavilov Institute of Plant Genetic Resources) or consider simplifying the title.
Answer
The title has been revised according to the comments.
Comments:
The abstract should be rewritten to include more quantitative data. Clearly state the objective, methods, main findings, and practical significance in a concise way. Please avoid overly general sentences and emphasize the main outcomes and how they advance knowledge in functional bakery formulations.
Answer
The abstract was rewritten according to the comments.
Comments:
The discussion section would benefit from a stronger comparison with previous studies, especially regarding biochemical composition and baking performance. Include references to similar studies on composite flour breads. This will help contextualize your findings and highlight the novelty of your work.
Answer
We have expanded the discussion of the results, comparing the current findings with our previous results (2020-21) and with data from other researchers. Unfortunately, there are few such works.
We have also provided a summary of the results for each section (biochemical and technological assessment) and highlighted the chufa samples that show potential for various branches of the food industry.
The correlation results of biochemical and technological assessments of flour and bread from different samples of chufa are presented in the statistical processing sections.
In a previous study, we assessed the biochemical properties of chufa grown in the Krasnodar and Tambov regions. We present a comparison of our previous data and current data.
The data from the current study are of a preliminary nature. Our primary goal was to evaluate the potential of chufa for bakery production. The next stage of research will involve assessing the baking and biochemical qualities of composite bread with different ratios of chufa tuber flour to wheat flour (40:60, 50:50, and 70:30), as well as studying flour from chufa tubers cultivated in other regions: the Central Black Earth and the North-Western regions of Russia.
Comments:
While descriptive statistics, ANOVA, and factor analysis are presented, it would help to clarify the number of replicates for each test and the confidence levels. Consider adding letters or symbols on graphs to indicate significant differences where applicable.
Answer
The number of biological replicates for each analysis was chosen by adapting the analytical procedure for the screening of a large sample set. The data obtained from the study of biological replicates are used for subsequent statistical analysis provided that the variation between replicates is within the instrument's technical error margin. Otherwise, additional biological replicates are analyzed until the required consistency is achieved. It should be noted that the regional regulatory documents recommend a minimum of two biological replicates.
Comments:
Figures 1 (A, B), 2 (A, B), 3 (A, B), and 4 (A, B) should be merged into single composite figures with clear sub-labels (A–B). This will improve readability and reduce redundancy.
Answer
Figures 1–5 were modified to reflect the recommendations, and the significance level was indicated in those figures where possible. Other significance data are presented in the text and highlighted in the table (correlation analysis).
Comments:
Although the study is comprehensive, the novelty statement should be emphasized in both the Introduction and Conclusion. Explicitly state how these findings can contribute to developing functional bread products or valorizing chufa tubers in the food industry.
Answer
The Introduction and Conclusion have been revised according to the comments.
Comments:
Please expand the Conclusion slightly to summarize main findings quantitatively, note key chufa accessions with superior performance, and suggest potential industrial applications.
Answer
The Conclusion has been expanded and revised according to the comments.
Comments:
Some minor editing in English is required. Several sentences are long and could be simplified to improve clarity and flow.
Answer
The text has been revised according to the comments.
Minor Comments
Comments:
- Use periods, not commas, e.g. 6.15%, not 6,15%
Answer
The text has been revised according to the comments.
Comments:
- Define all abbreviations at first use
Answer
The text has been revised according to the comments.
Comments:
- Ensure unit consistency throughout the manuscript
Answer
The text has been revised according to the comments.
Comments:
- Consider adding “functional foods,” “composite flour,” and/or “technological evaluation” as keywords
Answer
The keywords have been revised according to the comments.
All corrections in the text are marked in green.

Round 2
Reviewer 1 Report
Comments and Suggestions for Authors
The paper has been revised according to the comments, and it is recommended for acceptance and publication.
Reviewer 2 Report
Comments and Suggestions for Authors
The quality of the manuscript has significantly improved following the revision. The authors have addressed all my suggestions, and therefore, I recommend that it be accepted for publication in its current form.